# POSTERIOR RESTORATION FOR ENHANCED LLM PRUNING

## ABSTRACT

Pruning compresses and accelerates deep neural networks, making it important for deploying large language models (LLMs). However, traditional pruning methods, which use prior criteria in dense models to evaluate weight importance, face two key limitations: (1) they often overfit to calibration data and (2) they ignore weight interactions, leading to inaccurate importance estimation. To address these limitations, we propose posterior restoration, a simple two-stage approach. First, we apply a conventional prior criterion to generate an initial coarse pruning mask. Second, we restore the most important weights, guided by our novel posterior criteria (magnitude, global, and local), which re-evaluate the removed weights from the perspective of the already-pruned model. This unique viewpoint mitigates overfitting and captures previously ignored weight interactions. A key advantage of this scheme is its ability to seamlessly integrate and enhance most existing pruning methods. Experiments on Llama-3.1-8B and Mistral-7B across unstructured, channel-wise, and 2:4 sparsity patterns demonstrate that posterior restoration generally enhances pruned model performance. Our results show that the data-independent posterior magnitude criterion effectively mitigates overfitting, while the posterior global and local criteria successfully capture weight interactions.

## 1 INTRODUCTION

Pruning Ling et al. (2024); Ma et al. (2023); Frantar & Alistarh (2023) is an efficient method to reduce LLM inference costs by removing redundant weight elements. Because not all weight elements are essential for correct inference in LLMs, pruning methods first estimate the importance of weight elements, then remove the less important ones to reduce parameter size and thereby decrease inference costs. However, accurately estimating weight importance remains a fundamental challenge in LLM pruning.

Previous pruning methods assess weight importance through prior criteria Han et al. (2015); Dong et al. (2024); van der Ouderaa et al. (2024), which estimate importance based on information from the dense LLM. In other words, these methods treat the dense LLM as a prior and approximate the pruning loss of removing specific weight elements based on this prior. However, prior criteria face two key limitations: (1) overfitting to calibration data and (2) omitting interactions between weights to be pruned. Some prior criteria are designed to minimize inference loss on sampled calibration data, which may cause overfitting problems Sun et al. (2024); Molchanov et al. (2019). Meanwhile, prior criteria must omit weight interactions to ensure practical time complexity for importance estimation, which sacrifices estimation accuracy LeCun et al. (1989); Molchanov et al. (2019).

We propose posterior restoration to address these two limitations. The key insight of posterior restoration is to refine the pruning mask by restoring important weights based on observations of the pruned LLM. On one hand, we can re-estimate the importance of pruned weights without using calibration data and restore the more important ones to alleviate overfitting. On the other hand, we can optimize the pruning mask to minimize inference loss on calibration data based on the pruned LLM, thereby capturing interactions between pruned weights. However, implementing posterior restoration presents two challenges: (1) how to generate the initial pruning mask before applying

Figure 1: The overview of the proposed two-stage pruning scheme. The grey and white blocks denote the nonzero and zero elements, respectively. Meanwhile, the blue and green blocks denote the importance scores calculated through the prior and posterior criteria, respectively.

posterior restoration and (2) how to estimate the importance of pruned weights based on the pruned LLM.

We present a two-stage pruning scheme and posterior criteria to implement posterior restoration as shown in Fig. 1. Specifically, we first generate a coarse pruning mask using prior criteria, then refine the pruning mask by restoring pruned weights based on posterior criteria, which estimate the importance of the pruned weights. We present three types of posterior criteria: magnitude, global, and local, to address the limitations of prior criteria. Specifically, the posterior magnitude criterion can alleviate overfitting to calibration data, while the posterior global and local criteria are able to capture interactions between pruned weights for more accurate importance estimation.

We evaluated the proposed posterior restoration and pruning scheme on Llama-3.1-8B Touvron et al. (2023) and Mistral-7B Jiang et al. (2023) using various benchmarks. We selected WikiText2 Merity et al. (2016) as the calibration dataset and evaluate zero-shot generalization accuracy on several downstream tasks (ARC Clark et al. (2018), MMLU Hendrycks et al. (2020), etc.). In other words, for pruned LLMs, we evaluated fine-tuned inference accuracy on WikiText2 and generalization ability on the other dataset. The results show that the proposed posterior restoration can improve both fine-tuned inference accuracy and generalization ability compared with previous pruning methods.

## 2 RELATED WORKS

DNN pruning has been studied since the 1980s, where LeCun et al. (1989) proposed Optimal Brain Damage (OBD) to remove weight elements while minimizing pruning loss. The workflow of DNN pruning was then established as three steps: (1) estimating weight importance Zhang et al. (2024); Dong et al. (2024); Molchanov et al. (2019), (2) removing weights in a specific sparsity pattern Li & Ino (2024); Hu et al. (2024); Park et al. (2023), and (3) updating remaining weights for performance recovery Meng et al. (2024); Lin et al. (2024); Kim & Yoo (2025). As we define that previous pruning methods are based on prior criteria to estimate weight importance, we denote the weight importance estimation and weight removal steps as the prior pruning stage. To enhance this workflow, we integrate the posterior restoration between the prior pruning and the weight updating stages.

We derive the proposed posterior criteria based on the previous prior criteria, which can be classified into three categories: magnitude, global, and local. The magnitude criterion Gupta et al. (2024); Li et al. (2024) is the most straightforward one because it only focuses on the weight magnitude itself. The global criterion Molchanov et al. (2019); van der Ouderaa et al. (2024) focuses on minimizing the error of the pruned model on the final output. Meanwhile, the local criterion Frantar & Alistarh (2023); van der Ouderaa et al. (2024) simplifies the optimization problem in the global criterion by minimizing the pruning error on each layer of the DNN. Therefore, the local criterion is also called the "layer-wise" criterion.

An alternative solution for addressing the challenge of omitting interactions is iterative pruning Huang et al. (2025); Wu et al. (2024); Zhang et al. (2021), which repeats the process of pruning and gradually increase the pruning ratio. The iterative pruning scheme also captures the interactions of pruned weights because the importance of the remaining weights is calculated based on the pruned model. However, the proposed posterior restoration achieves better performance for pruned

LLMs than the iterative pruning scheme in our experiments, indicating that posterior restoration is more effective than the previous iterative pruning approach.

Restoring pruned weights during the pruning process is a common approach Cho et al. (2023); Hoefler et al. (2021). However, this weight restoration strategy is applied in a preliminary way with sparse training Zhou et al. (2021); Huang et al. (2025), lottery ticket-based pruning Frankle & Carbin (2018); Yue et al. (2025), and Bayesian neural networks Wright et al. (2024). To the best of our knowledge, we are the first to apply the weight restoration strategy to the criterion-based post-training pruning area and propose importance criteria for estimating the importance of pruned weights.

## 3 PRELIMINARIES

To explain posterior restoration in detail, we introduce prior criteria for estimating weight importance and define our notation. Prior criteria minimize the pruning loss by estimating how pruning degrades the inference accuracy of LLMs.

An LLM with $l$ layers is composed of individual layer matrices $\{W_1, ..., W_l\}$. For convenience, we treat these as a single flattened vector, $\mathbf{W}$. Pruning is performed by applying binary masks $\{M_1, ..., M_l\}$ for the corresponding layers, which are also flatten these as a vector $\mathbf{M}$. The pruned weights are thus defined via a Hadamard product as $\mathbf{W}' = \mathbf{W} \odot \mathbf{M}$.

The goal of pruning is to find an optimal mask $\mathbf{M}$ that minimizes the degradation in model performance under a specific sparsity constraint. Letting $L(X; \mathbf{W})$ be the inference loss on an input $X$, this optimization objective is formulated as:

$$\arg\min_{\mathbf{M}} \quad |L(X; \mathbf{W}) - L(X; \mathbf{W}')|$$
$$\text{subject to} \quad \frac{\text{zeros}(\mathbf{M})}{\text{size}(\mathbf{M})} = p, \tag{1}$$

where $\text{zeros}(\cdot)$ and $\text{size}(\cdot)$ count the number of zero elements and the total number of elements in the mask, respectively, and $p$ is the target pruning ratio.

Brute-force calculation of the optimal pruning mask for $\mathbf{W}'$ is difficult. Therefore, various prior criteria are proposed to approximate the pruning loss caused by removing a specific weight elements. In this paper, we use $\mathcal{I}(w)$ to denote the criterion function for calculating the importance score of the weight element $w$. To further simplify the calculation, we can assume that:

**Assumption 1** *The pruning error incurred by removing each weight element is independent of each other.*

The objective function of Eq. (1) is then simplified as follows:

$$\arg\min_{\mathbf{M}} \sum_{w \in (\mathbf{W} \odot \neg \mathbf{M})} \mathcal{I}(w)$$
$$\text{subject to} \quad \text{zeros}(\mathbf{M})/\text{size}(\mathbf{M}) = p. \tag{2}$$

Note that we flip the mask $M$ in Eq. (2) because $\mathcal{I}(w)$ gives the error caused by removing $w$.

We classify the previous prior criteria into three categories based on their optimization focus: magnitude, global, and local. The magnitude criterion focuses on individual weights, the global criterion focuses on the final output, and the local criterion focuses on each layer's output.

### 3.1 MAGNITUDE CRITERION

For the magnitude criterion, we assume that weight importance scores are independent of the inputs. Therefore, we have:

**Assumption 2** *The input of each layer follows a uniform distribution.*

Under Assumption 2, weight elements with larger magnitudes are more important because all inputs are identical. This assumption is straightforward and reasonable because removing large weights is

more likely to change the LLM output significantly compared with pruning small weights. Consequently, the prior magnitude criterion $\mathcal{I}_m$ can be defined as follows:

$$\mathcal{I}_m(w) = w^2. \tag{3}$$

The importance criterion of Eq. (3) is also called the L2 criterion Huang et al. (2021) and represents one typical magnitude criterion. We select the L2 criterion as our baseline method because researchers have shown that L2 pruning is the most general magnitude pruning method Huang et al. (2021).

The magnitude criterion is easy to implement and general because it does not depend on input calibration data. However, the magnitude criterion may lead to poor inference accuracy when the input distribution has large variance.

## 3.2 GLOBAL CRITERION

Unlike the magnitude criterion, the global criterion aims to minimize the pruning loss on a specific dataset. We define the pruning loss as $\mathcal{L} = |L(X; \mathbf{W}) - L(X; \mathbf{W}')|$. This term can be approximated by a second-order Taylor expansion of the loss function $L(X; \mathbf{W}')$ around the dense weights $\mathbf{W}$, which provides the following estimate:

$$\mathcal{L} \approx |(\mathbf{W}' - \mathbf{W})^\top \cdot \frac{\partial L(X; \mathbf{W})}{\partial \mathbf{W}} + \frac{1}{2}(\mathbf{W}' - \mathbf{W})^\top \cdot \frac{\partial^2 L(X; \mathbf{W})}{(\partial \mathbf{W})^2} \cdot (\mathbf{W}' - \mathbf{W})| \tag{4}$$

Under the independence assumption (Assumption 1), this total loss is treated as the sum of changes from removing each individual weight $w$. Because for each pruned weight, the change is $(0 - w) = -w$, the expression simplifies to:

$$\mathcal{L} \approx \sum_{w \in (\mathbf{W} \odot \neg \mathbf{M})} |-w \cdot \frac{\partial L(X; \mathbf{W})}{\partial w} + \frac{w^2}{2} \cdot \frac{\partial^2 L(X; \mathbf{W})}{(\partial w)^2}|. \tag{5}$$

To further simplify Eq. (5), we select Molchanov et al. (2019)'s method, which omits the Jacobian and approximates the Hessian through Fisher Approximation Hoefler et al. (2021). Consequently, the prior global criterion $\mathcal{I}_g$ is then

$$\mathcal{I}_g(w, \frac{\partial L(X; \mathbf{W})}{\partial w}) = (w \cdot \frac{\partial L(X; \mathbf{W})}{\partial w})^2. \tag{6}$$

Compared with the magnitude criterion, the global criterion can capture properties of the input distribution, achieving smaller pruning loss on the calibration dataset. However, the global criterion tends to overfit to the calibration dataset, leading to larger pruning loss for inputs outside the calibration dataset.

## 3.3 LOCAL CRITERION

The local criterion estimates pruning loss more accurately than the global criterion. Instead of minimizing the pruning error on the final layer output, the local criterion minimizes the pruning error on each layer's output. Specifically, the local criterion assumes:

**Assumption 3** *The pruning loss is minimized if the error on each layer's output is minimized.*

Specifically, the pruning loss for the $i$-th layer is:

$$\mathcal{L}_i = \|I_i W_i^\top - I_i'(W_i \odot M_i)^\top\|_F, \tag{7}$$

where $I_i$ is the output matrix from the dense $(i-1)$-th layer, $I_i'$ is the input from the pruned $(i-1)$-th layer, and $\|\cdot\|_F$ denotes the Frobenius norm. Note that the actual input for each layer may differ from the dense model because previous layers are pruned and produce altered inputs.

To simplify the optimization problem, we further assume:

**Assumption 4** *The pruning error on each layer can be ignored. In other words, $I_i \approx I_i'$.*

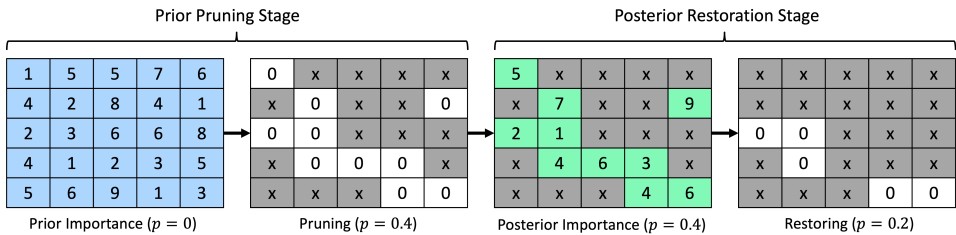

Figure 2: Example of the two-stage pruning scheme. We show a $5 \times 5$ weight matrix, where each cell denotes one weight element. The colors denote different statuses of the weights: blue for prior importance scores, green for posterior importance scores, grey for preserved weights, and white for pruned weights.

Note that Assumption 4 is a variant of Assumption 1 because Assumption 4 considers the layers in the model independently. According to Assumptions 3 and 4, we define the pruning loss under the local criterion:

$$\mathcal{L} \approx \sum_{(W_i \odot M_i) \in \mathbf{W}'} \| I_i W_i^\top - I_i (W_i \odot M_i)^\top \|_F. \tag{8}$$

Previous studies Frantar & Alistarh (2023); van der Ouderaa et al. (2024); Hassibi et al. (1993) have proposed various solutions for simplifying Eq. (8). We select Wanda Sun et al. (2024) as our baseline because Wanda is simple yet effective compared with its counterparts. In Wanda, the local criterion $\mathcal{I}_l$ derived from Eq. (8) is given as:

$$\mathcal{I}_l(w_{x,y}, I_i) = (w_{x,y} \cdot \| I_i^{:,y} \|_F)^2, \tag{9}$$

where $w_{x,y}$ is the weight element at the $x$-th row and $y$-th column and $I_{:,y}$ denotes the $y$-th column vector in $I_i$.

Similar to the global criterion, the local criterion also suffers from overfitting because Eq. (9) only minimizes the pruning loss on sampled inputs from the calibration dataset.

## 4 METHOD

We propose posterior restoration to address two key limitations of prior pruning criteria: overfitting and the omission of weight interactions. First, criteria that depend on calibration data, such as global and local methods, can overfit by optimizing the pruning loss on a small, specific dataset. Second, all prior criteria estimate the importance of each weight independently, which simplifies the computation but ignores the potential interactions between weights.

Our method, posterior restoration, has two main components: a two-stage pruning scheme and a set of posterior criteria. The key insight is to first apply a standard pruning method to create an initial coarse mask. We then refine this mask by restoring the most important weights that were just removed. This restoration process is guided by our posterior criteria, which are designed to overcome the limitations of prior methods by re-evaluating pruned weights, either without relying on calibration data or by explicitly considering their interactions.

### 4.1 TWO-STAGE PRUNING SCHEME

The two-stage pruning scheme provides the framework for posterior restoration. This resolves a fundamental prerequisite: the restoration step can only be applied to a model that has already undergone pruning. Our scheme addresses this by operating in two stages. First, we apply a prior criterion to generate an initial coarse pruning mask that is sparser than our final target. Second, we use our posterior criteria to refine this mask by restoring the most important pruned weights until we reach the desired sparsity level.

To achieve a final pruning ratio of $p$, we first create a coarse mask $\mathbf{M}$ by pruning the model to a higher ratio of $p + \Delta p$. Here, $\Delta p \in (0, 1 - p)$ is a hyperparameter we call the **restoring ratio**. The subsequent posterior restoration stage then aims to refine this initial mask into a new mask, $\mathbf{M}_r$, that

minimizes the pruning loss by restoring a subset of the previously removed weights. The process seeks a refined mask $\mathbf{M}_r$ such that $p = \text{zeros}(\mathbf{M}_r)/\text{size}(\mathbf{M}_r)$, derived from the initial mask where $p + \Delta p = \text{zeros}(\mathbf{M})/\text{size}(\mathbf{M})$. The formal optimization objective is given below:

$$\arg\min_{\mathbf{M}_r} \quad |L(X; \mathbf{W}) - L(X; \mathbf{W}'_r)|$$
$$\text{subject to} \quad \text{zeros}(\mathbf{M}_r)/\text{size}(\mathbf{M}_r) = p, \tag{10}$$
$$\text{supp}(\mathbf{M}) \subseteq \text{supp}(\mathbf{M}_r),$$

where we use $\text{supp}(\cdot)$ to denote the support of a mask, i.e., the set of indices corresponding to its non-zero (preserved) elements.

The procedural steps for our two-stage pruning scheme are detailed in Algorithm 1. The algorithm consists of two main phases: a prior pruning stage (Lines 1–2) to generate an initial coarse mask, and a posterior restoration stage (Lines 3–4) to refine this mask.

---

**Algorithm 1** Two-stage pruning

---

**Require:** Weights $\mathbf{W}$, calibration dataset $\mathbb{X}$, final pruning ratio $p$, and restoring ratio $\Delta p$.
**Return:** Refined pruning mask $\mathbf{M}_r$.
1: Calculate the prior importance scores for $\mathbf{W}$ (using $\mathbb{X}$ if required).
2: Generate coarse mask $\mathbf{M}$ by pruning weights with the lowest prior importance to achieve a pruning ratio of $p + \Delta p$.
3: Calculate the posterior importance scores for the weights pruned by $\mathbf{M}$ (using $\mathbb{X}$ if required).
4: Generate refined mask $\mathbf{M}_r$ by restoring the $\Delta p$ fraction of pruned weights with the highest posterior importance scores.

---

Figure 2 provides a visual example of our two-stage pruning scheme, illustrating the process of achieving a final pruning ratio of $p = 0.2$ with a restoring ratio $\Delta p = 0.2$. The initial dense weight matrix (leftmost panel, with $p = 0$) has importance scores calculated by a prior criterion. In the prior pruning stage, we remove 40% of the least important weights to create a coarse pruning mask (second panel, $p = 0.4$). For the posterior restoration stage, we then estimate the posterior importance scores for the weights in this coarse mask. Finally, we restore 20% of the most important previously pruned weights to generate the refined pruning mask (rightmost panel, $p = 0.2$).

## 4.2 Posterior Criteria

We introduce three posterior criteria to re-evaluate the importance of weights within the coarsely pruned model. Each criterion is designed to overcome a specific limitation of its prior counterpart described in the Preliminary section. The posterior magnitude criterion mitigates the overfitting introduced by data-dependent prior methods (i.e., global and local). Conversely, the posterior global and local criteria are designed to capture the weight interactions that are ignored under the independence assumption of all prior methods.

### 4.2.1 Posterior Magnitude Criterion

The posterior magnitude criterion aims to mitigate the overfitting observed in data-dependent prior methods. To achieve this, we re-evaluate pruned weights without relying on a calibration dataset, thereby adopting the core principle of the prior magnitude criterion for the restoration stage. We therefore extend Assumption 2 for the posterior setting:

**Assumption 5** *When data is given out of the calibration dataset, the input of each layer follows a uniform distribution.*

According to Assumption 5, we define the posterior magnitude criterion $\mathcal{I}_m^r$ as:

$$\mathcal{I}_m^r(w) = w^2. \tag{11}$$

While this formula is identical to its prior counterpart, its power lies in its application. The prior magnitude criterion is used in the first stage to help prune a dense model. In contrast, the posterior version is applied during the second stage to re-evaluate and restore weights that were pruned by a potentially overfitting, data-dependent criterion. By providing this data-independent "second opinion," it counteracts the overfitting introduced in the first stage, a finding our experimental results confirm.

### 4.2.2 Posterior Global Criterion

The posterior global criterion is designed to capture the weight interactions that prior methods ignore by reversing the perspective of the Taylor expansion.

Specifically, to estimate the pruning error, the prior criterion approximates the loss of the sparse model by expanding $\mathbf{L}(\mathbf{X}; \mathbf{W}')$ around the dense weights $\mathbf{W}$ in Eq. (4). In contrast, to estimate the benefit of restoring, our posterior method approximates the loss of the dense model by expanding $\mathbf{L}(\mathbf{X}; \mathbf{W})$ around the sparse weights $\mathbf{W}'$. Applying this logic, the approximation for the change in loss becomes:

$$\mathcal{L} \approx \sum_{w \in (\mathbf{W} \odot \neg(\mathbf{M}_r))} |w \cdot \frac{\partial L(X; \mathbf{W}')}{\partial w} + \frac{w^2}{2} \cdot \frac{\partial^2 L(X; \mathbf{W}')}{(\partial w)^2}|, \tag{12}$$

The crucial difference is that the gradients are now evaluated on the pruned model with $\mathbf{W}'$. Therefore, each weight's importance is assessed in the context of the sparse network, allowing the score to capture these interactive effects. Following an approximation similar to that of Molchanov et al. (2019), the posterior global criterion $\mathcal{I}_g^r$ is then defined as:

$$\mathcal{I}_g^r(w, \frac{\partial L(X; \mathbf{W}')}{\partial w}) = (w \cdot \frac{\partial L(X; \mathbf{W}')}{\partial w})^2. \tag{13}$$

### 4.2.3 Posterior Local Criterion

The posterior local criterion approaches the layer-wise optimization from a different direction than its prior counterpart. The prior criterion simplifies its calculation of the layer-wise loss (Eq. (8)) by approximating a layer's true input from the sparse model, $I_i'$, with the input from the dense model, $I_i$ (Assumption 4). This allows for efficient estimation but ignores the error accumulated from previously pruned layers.

Our posterior method offers an alternative perspective. During the restoration stage, the true inputs from the pruned model, $I_i'$, are directly available. By using these inputs, the posterior criterion can account for the state of the pruned network. This core idea—using the sparse model's state to re-evaluate weights—is the same principle behind the posterior global criterion. This leads to a re-formulation of the layer-wise pruning loss (Eq. (7)):

$$\mathcal{L} \approx \sum_{(W_i \odot M_{r_i}) \in \mathbf{W}'} \|I_i' W_i^\top - I_i'(W_i \odot M_i)^\top\|_F. \tag{14}$$

This objective minimizes the layer's output error given the actual input it receives from the preceding pruned layer. By adapting the Wanda simplification (Eq.(9)) to this alternative loss formulation, we define the posterior local criterion $\mathcal{I}_l^r$ as:

$$\mathcal{I}_l^r(w_{x,y}, I_i') = (w_{x,y} \cdot \|I_i'^{:,y}\|_F)^2. \tag{15}$$

## 5 Experiments

We conduct a series of experiments on modern LLMs to evaluate the effectiveness of posterior restoration. Our evaluation is designed to answer three key questions: **(1)** Does our method improve performance across various sparsity patterns? **(2)** How does the restoring ratio, $\Delta p$, affect the outcome? **(3)** How does posterior restoration compare to conventional iterative pruning? Our results demonstrate that while our method generally enhances model performance, the optimal combination of prior and posterior criteria depends on the specific pruning setting.

### 5.1 Experiment Settings

Our experiments are conducted on Llama-3.1-8B Touvron et al. (2023) and Mistral-7B Jiang et al. (2023), selected for their general architectures. Following prior work van der Ouderaa et al. (2024); Ma et al. (2023), we use WikiText2 Merity et al. (2016) as our calibration dataset. To evaluate generalization and test for overfitting, we measure zero-shot accuracy on a suite of downstream tasks (ARC Clark et al. (2018), BoolQ Clark et al. (2019), Hellaswag Zellers et al. (2019), and

Table 1: Perplexities and inference accuracies of pruned Llama-3.1-8B across unstructured, channel-wise, and 2:4 sparsity. The best results for each pattern are highlighted in bold.

| Sparse Pattern | Pruning Ratio | Stage I Criterion | Stage II Criterion | $\Delta p$ | PPL | Avg. Others |
|---|---|---|---|---|---|---|
| dense | 0% | N/A | N/A | N/A | 8.64 | 68.93 |
| unstructured | 50% | $\mathcal{I}_l$ | None | 0 | 17.75 | 58.88 |
| | | | $\mathcal{I}_m^r$ | 0.1 | **16.76** | **59.90** |
| | | | $\mathcal{I}_g^r$ | 0.1 | 19.51 | 47.11 |
| | | | $\mathcal{I}_l^r$ | 0.1 | 20.60 | 48.38 |
| | | | $\mathcal{I}_m$ | -0.1 | 29.93 | 53.99 |
| | | | $\mathcal{I}_g$ | -0.1 | 90.46 | 37.54 |
| | | | $\mathcal{I}_l$ | -0.1 | 17.67 | 57.92 |
| | 30% | $\mathcal{I}_l$ | None | 0 | 11.84 | 67.62 |
| | | $\mathcal{I}_m$ | $\mathcal{I}_l^r$ | 0.1 | **11.62** | **68.05** |
| channel-wise | 10% | $\mathcal{I}_g$ | None | 0 | 26.02 | 47.59 |
| | | | $\mathcal{I}_m^r$ | 0.1 | 38.39 | 48.38 |
| | | | $\mathcal{I}_g^r$ | 0.1 | **22.67** | **52.16** |
| | | | $\mathcal{I}_l^r$ | 0.1 | 35.42 | 46.82 |
| 2:4 | 50% | $\mathcal{I}_l$ | None | 0 | **32.94** | 44.18 |
| | | | $\mathcal{I}_m^r$ | 0.25 | 33.85 | **46.62** |
| | | | $\mathcal{I}_g^r$ | 0.25 | 91.52 | 32.33 |
| | | | $\mathcal{I}_l^r$ | 0.25 | 182.07 | 30.86 |

MMLU Hendrycks et al. (2020)). For brevity, we report the average accuracy across these tasks as "Avg. Others." The main text presents results for Llama-3.1-8B. The full results for Mistral-7B, which show similar trends, as well as a detailed breakdown of the "Avg. Others" scores, are provided in the Appendix A.

We follow a standard post-training pruning methodology. We evaluate three sparsity patterns: unstructured (element-wise) Han et al. (2015), channel-wise (row) Ma et al. (2023), and 2:4 Hu et al. (2024). The same pruning ratio is applied across all layers, and we subsequently fine-tune each model on WikiText2 using Low-Rank Adaptation (LoRA) Hu et al. (2022) to recover performance. To simplify the discussion, we denote our two-stage pruning configurations using the format `Prior + Posterior`. For example, the notation $\mathcal{I}_l + \mathcal{I}_m^r$ refers to a setup using the prior local criterion ($\mathcal{I}_l$) in Stage I and the posterior magnitude criterion ($\mathcal{I}_m^r$) in Stage II.

### 5.2 FIXED RESTORING RATIO

In our first experiment, we evaluate posterior restoration across the three sparsity patterns using a fixed restoring ratio ($\Delta p$), with results presented in Table 1. We set $\Delta p = 0.1$ for the unstructured and channel-wise patterns. For the 2:4 pattern, we set $\Delta p = 0.25$. This value is chosen because our process first prunes the model to a 1:4 sparsity (one non-zero weight per block of four), and then restores one weight in each block to reach the final 2:4 structure. To maintain focus, Table 1 reports the results for the best-performing prior criterion for each sparsity pattern; full results are available in the Appendix A.

The results in Table 1 show that the posterior magnitude criterion ($\mathcal{I}_m^r$) is highly effective at improving generalization accuracy, particularly when paired with the data-dependent local prior ($\mathcal{I}_l$). For unstructured sparsity, the $\mathcal{I}_l + \mathcal{I}_m^r$ combination improves upon the baseline in both perplexity (17.75 → 16.76) and average accuracy (58.88 → 59.90). It also boosts accuracy for 2:4 sparsity (44.18 → 46.62). These findings support our hypothesis that a data-independent posterior criterion can counteract the overfitting introduced by a data-dependent prior.

The posterior global criterion ($\mathcal{I}_g^r$) shows a highly specialized effect, yielding significant gains for coarse-grained sparsity but degrading performance for fine-grained patterns. As shown in Table 1, the $\mathcal{I}_g + \mathcal{I}_g^r$ combination provides a substantial improvement for the channel-wise pattern, reducing perplexity (26.02 → 22.67) and boosting average accuracy (47.59 → 52.16). Conversely, for both unstructured and 2:4 sparsity, this criterion harms performance across both metrics. This suggests

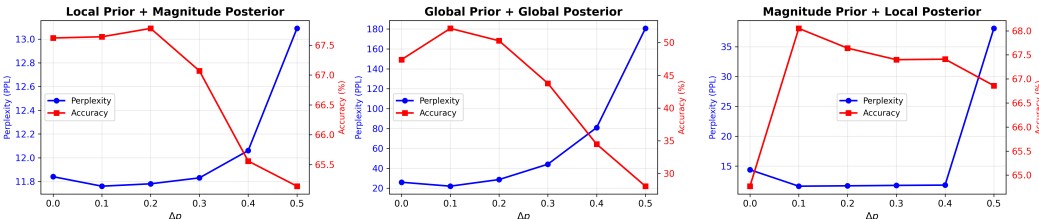

Figure 3: The perpleixties and average other accuracies on Llama-3.1-8B with $\Delta p$ across 0 to 0.5. Each panel shows the results of corresponding pruning setting on the title. The blue lines represent the perplexities while the red ones represent the average other accuracies.

that the posterior global criterion is particularly suited for structured pruning, where capturing the interactions between large blocks of weights, such as entire channels, is critical.

The posterior local criterion ($\mathcal{I}_l^r$) is most effective when paired with the magnitude prior ($\mathcal{I}_m$). For instance, as shown in Table 1 for unstructured pruning at 30% sparsity, the $\mathcal{I}_m + \mathcal{I}_l^r$ combination achieves the best performance, improving both perplexity and average accuracy.

To compare our method against a conventional iterative pruning baseline, we conduct a control experiment with $\Delta p = -0.1$ for the unstructured pattern. This denotes a two-step pruning process where the model is first pruned to a sparsity of $p - 0.1$, and then further pruned to the final sparsity $p$. Prior criteria are used for both steps. For this setup, the "Stage II Criterion" column in Table 1 denotes the prior criterion used for the second pruning step. The results demonstrate that posterior restoration yields better performance than this iterative pruning approach.

### 5.3 CHANGING RESTORING RATIO

In our second experiment, we investigate the impact of the restoring ratio, $\Delta p$, to identify its optimal range. We vary $\Delta p$ from 0 to 0.5 for the most effective prior-posterior combinations identified in our first experiment: $\mathcal{I}_l + \mathcal{I}_m^r$ for unstructured sparsity, $\mathcal{I}_g + \mathcal{I}_g^r$ for channel-wise sparsity at $p = 0.1$, and $\mathcal{I}_m + \mathcal{I}_l^r$ for unstructured sparsity at $p = 0.3$.

The results, plotted in Figure 3, show a clear trend across all three settings. Performance on both perplexity and average accuracy generally improves as $\Delta p$ increases from 0, peaking in the range of 0.1 to 0.2. Beyond this point, performance sharply degrades. This suggests that while a modest restoration is beneficial, pruning too aggressively in the first stage (i.e., a large $\Delta p$) damages the model to a point where the sparse network's information becomes unreliable for guiding an effective restoration.

## 6 CONCLUSION

In this paper, we introduced posterior restoration, a novel technique to enhance LLM pruning. Our method addresses two key limitations of existing approaches: (1) overfitting to calibration data and (2) neglecting weight interdependencies. We propose a two-stage pruning scheme that seamlessly integrates this restoration process with established pruning methods. The restoration is guided by three posterior criteria—magnitude, global, and local—for identifying and restoring the most important pruned weights. Our experimental results demonstrate that posterior restoration effectively improves the performance of pruned LLMs across various models and benchmarks.

Looking ahead, we can extend posterior restoration beyond one-shot pruning. It can be incorporated into conventional iterative pruning and sparse training frameworks. For instance, to achieve a 50% sparsity target, one could perform five steps of pruning 20% of the weights while restoring 10% using our method. Furthermore, posterior restoration could serve as a more effective alternative to the weight regrowth mechanisms in sparse training. We leave these promising directions for future work.

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

# A  APPENDIX

This appendix provides a detailed breakdown of our experimental results. The tables are grouped by model (Llama-3.1-8B, Mistral-7B) and sparsity pattern. Within each block of experiments for a specific pruning ratio, the best-performing result for each metric is highlighted in bold. Note that we omit results for the $\mathcal{I}_m + \mathcal{I}_m^r$ combination, as it is equivalent to the one-shot $\mathcal{I}_m$ baseline.

A key observation across all experiments is that for any given setting, a prior-posterior combination generally exists that outperforms the corresponding prior-only baseline. However, no single combination consistently performs the best across all scenarios. This suggests a valuable direction for future work: developing methods to automatically select the optimal criteria combination for a given model and sparsity target.

## A.1  LLAMA-3.1-8B

### A.1.1  UNSTRUCTURED

Table 2: Performance for Llama-3.1-8B with Unstructured sparsity.

| Pruning Ratio | Stage I Criterion | Stage II Criterion | Restoring Ratio | WikiText2 | Arc Challenge | Arc Easy | BoolQ | Hellaswag | MMLU | Average |
|---|---|---|---|---|---|---|---|---|---|---|
| 0.3 | $\mathcal{I}_g$ | None | 0.0 | 24.4988 | 34.39 | 66.62 | 70.73 | 44.17 | 46.09 | 52.40 |
| 0.3 | $\mathcal{I}_m$ | None | 0.0 | 14.3650 | 49.23 | 78.83 | 76.12 | 55.93 | 63.78 | 64.78 |
| 0.3 | $\mathcal{I}_l$ | None | 0.0 | 11.8400 | 51.79 | 80.18 | 83.09 | **58.03** | 65.05 | 67.62 |
| 0.3 | $\mathcal{I}_g$ | $\mathcal{I}_m^r$ | 0.1 | 12.1201 | 50.17 | 80.22 | 83.82 | 57.04 | 64.83 | 67.22 |
| 0.3 | $\mathcal{I}_l$ | $\mathcal{I}_m^r$ | 0.1 | 11.7680 | 51.79 | 80.05 | 83.09 | 58.01 | 65.29 | 67.65 |
| 0.3 | $\mathcal{I}_g$ | $\mathcal{I}_g^r$ | 0.1 | 15.3629 | 40.19 | 74.62 | 76.33 | 50.16 | 56.21 | 59.50 |
| 0.3 | $\mathcal{I}_m$ | $\mathcal{I}_g^r$ | 0.1 | 11.7470 | 50.85 | **81.06** | 82.94 | 57.17 | 64.54 | 67.31 |
| 0.3 | $\mathcal{I}_l$ | $\mathcal{I}_g^r$ | 0.1 | **11.5251** | 50.43 | 79.88 | 83.76 | 57.58 | 64.63 | 67.26 |
| 0.3 | $\mathcal{I}_g$ | $\mathcal{I}_l^r$ | 0.1 | 12.8677 | 45.90 | 78.54 | 83.85 | 55.73 | 62.61 | 65.33 |
| 0.3 | $\mathcal{I}_m$ | $\mathcal{I}_l^r$ | 0.1 | 11.6214 | 52.13 | 80.30 | **84.37** | 58.01 | **65.48** | **68.06** |
| 0.3 | $\mathcal{I}_l$ | $\mathcal{I}_l^r$ | 0.1 | 11.7435 | **52.22** | 80.30 | 80.67 | 57.96 | 65.26 | 67.28 |
| 0.5 | $\mathcal{I}_g$ | None | 0.0 | 294.8311 | 18.09 | 34.01 | 49.82 | 27.38 | 22.92 | 30.44 |
| 0.5 | $\mathcal{I}_m$ | None | 0.0 | 89.2929 | 28.84 | 55.13 | 67.28 | 38.69 | 45.11 | 47.01 |
| 0.5 | $\mathcal{I}_l$ | None | 0.0 | 17.7567 | **42.49** | 70.83 | 80.67 | 50.06 | 50.33 | 58.88 |
| 0.5 | $\mathcal{I}_g$ | $\mathcal{I}_m^r$ | 0.1 | 28.3283 | 35.41 | 68.98 | 68.87 | 44.44 | 45.22 | 52.58 |
| 0.5 | $\mathcal{I}_l$ | $\mathcal{I}_m^r$ | 0.1 | **16.7641** | 42.15 | **72.47** | 79.82 | **51.73** | **53.35** | **59.90** |
| 0.5 | $\mathcal{I}_g$ | $\mathcal{I}_g^r$ | 0.1 | 130.4709 | 17.32 | 38.59 | 59.51 | 28.25 | 23.07 | 33.35 |
| 0.5 | $\mathcal{I}_m$ | $\mathcal{I}_g^r$ | 0.1 | 23.1966 | 37.46 | 69.57 | 77.98 | 46.28 | 50.00 | 56.26 |
| 0.5 | $\mathcal{I}_l$ | $\mathcal{I}_g^r$ | 0.1 | 19.5072 | 35.92 | 67.97 | 73.15 | 46.53 | 48.20 | 54.35 |
| 0.5 | $\mathcal{I}_g$ | $\mathcal{I}_l^r$ | 0.1 | 82.2520 | 20.31 | 49.62 | 62.51 | 30.94 | 30.23 | 38.72 |
| 0.5 | $\mathcal{I}_m$ | $\mathcal{I}_l^r$ | 0.1 | 18.9486 | 39.16 | 70.29 | **81.74** | 48.38 | 50.78 | 58.07 |
| 0.5 | $\mathcal{I}_l$ | $\mathcal{I}_l^r$ | 0.1 | 20.6064 | 37.88 | 68.98 | 75.26 | 47.91 | 49.79 | 55.96 |
| 0.5 | $\mathcal{I}_g$ | $\mathcal{I}_m$ | -0.1 | 109.5595 | 20.56 | 47.47 | 62.32 | 31.12 | 25.37 | 37.37 |
| 0.5 | $\mathcal{I}_l$ | $\mathcal{I}_m$ | -0.1 | 25.1179 | 34.39 | 65.36 | 70.61 | 47.08 | 48.11 | 53.11 |
| 0.5 | $\mathcal{I}_g$ | $\mathcal{I}_g$ | -0.1 | 1600.8142 | 16.89 | 33.42 | 60.06 | 27.55 | 23.25 | 32.23 |
| 0.5 | $\mathcal{I}_m$ | $\mathcal{I}_g$ | -0.1 | 81.5360 | 21.33 | 44.99 | 69.36 | 34.04 | 31.33 | 38.97 |
| 0.5 | $\mathcal{I}_l$ | $\mathcal{I}_g$ | -0.1 | 42.9505 | 27.23 | 56.48 | 69.36 | 36.05 | 33.63 | 44.75 |
| 0.5 | $\mathcal{I}_g$ | $\mathcal{I}_l$ | -0.1 | 90.4680 | 20.90 | 48.91 | 63.82 | 30.83 | 23.25 | 34.54 |
| 0.5 | $\mathcal{I}_m$ | $\mathcal{I}_l$ | -0.1 | 29.9330 | 38.82 | 68.35 | 67.19 | 45.97 | 49.65 | 54.00 |
| 0.5 | $\mathcal{I}_l$ | $\mathcal{I}_l$ | -0.1 | 17.6763 | 41.21 | 71.00 | 77.61 | 49.89 | 49.90 | 57.92 |
| 0.7 | $\mathcal{I}_g$ | None | 0.0 | 1286.1294 | 18.77 | 27.36 | 37.83 | 26.43 | 22.95 | 26.67 |
| 0.7 | $\mathcal{I}_m$ | None | 0.0 | 47343.1318 | 18.94 | 27.74 | 37.86 | 26.28 | 22.94 | 26.75 |
| 0.7 | $\mathcal{I}_l$ | None | 0.0 | **228.7446** | **20.05** | 36.20 | **56.67** | **28.05** | 23.20 | **32.83** |
| 0.7 | $\mathcal{I}_g$ | $\mathcal{I}_m^r$ | 0.1 | 482.1695 | 19.62 | 32.62 | 39.20 | 27.17 | **24.42** | 28.61 |
| 0.7 | $\mathcal{I}_l$ | $\mathcal{I}_m^r$ | 0.1 | 271.8747 | 20.39 | **36.28** | 54.65 | 27.56 | 22.94 | 32.36 |
| 0.7 | $\mathcal{I}_g$ | $\mathcal{I}_g^r$ | 0.1 | 734.6019 | 18.09 | 28.66 | 37.83 | 26.48 | 22.91 | 26.79 |
| 0.7 | $\mathcal{I}_m$ | $\mathcal{I}_g^r$ | 0.1 | 3633.7460 | 18.26 | 30.26 | 37.83 | 26.88 | 22.94 | 27.23 |
| 0.7 | $\mathcal{I}_l$ | $\mathcal{I}_g^r$ | 0.1 | 412.7045 | 19.62 | 29.55 | 37.95 | 26.81 | 23.00 | 27.39 |
| 0.7 | $\mathcal{I}_g$ | $\mathcal{I}_l^r$ | 0.1 | 916.3088 | 18.43 | 28.87 | 37.83 | 26.37 | 22.92 | 26.88 |
| 0.7 | $\mathcal{I}_m$ | $\mathcal{I}_l^r$ | 0.1 | 1192.6996 | 19.71 | 28.49 | 37.98 | 26.47 | 22.96 | 27.12 |
| 0.7 | $\mathcal{I}_l$ | $\mathcal{I}_l^r$ | 0.1 | 440.2192 | 19.62 | 29.92 | 41.07 | 26.70 | 23.26 | 28.11 |

### A.1.2  CHANNEL-WISE

Table 3: Performance for Llama-3.1-8B with Channel-wise sparsity.

| Pruning Ratio | Stage I Criterion | Stage II Criterion | Restoring Ratio | WikiText2 | Arc Challenge | Arc Easy | BoolQ | Hellaswag | MMLU | Average |
|---|---|---|---|---|---|---|---|---|---|---|
| 0.1 | $\mathcal{I}_g$ | None | 0.0 | 26.0200 | 30.72 | 63.89 | 63.98 | 39.09 | 40.29 | 47.59 |
| 0.1 | $\mathcal{I}_m$ | None | 0.0 | 67606.6661 | 20.82 | 24.75 | 37.92 | 25.48 | 24.54 | 26.70 |
| 0.1 | $\mathcal{I}_l$ | None | 0.0 | 45921.8622 | 20.99 | 25.55 | 37.86 | 26.05 | 22.81 | 26.65 |
| 0.1 | $\mathcal{I}_g$ | $\mathcal{I}_m^r$ | 0.1 | 38.3933 | 30.63 | 62.88 | 72.29 | 37.20 | 38.93 | 48.39 |
| 0.1 | $\mathcal{I}_l$ | $\mathcal{I}_m^r$ | 0.1 | 37226.9855 | 20.56 | 25.25 | 37.80 | 25.98 | 22.94 | 26.51 |
| 0.1 | $\mathcal{I}_g$ | $\mathcal{I}_g^r$ | 0.1 | **22.6742** | **33.45** | **66.41** | **75.60** | **42.83** | **42.55** | **52.17** |
| 0.1 | $\mathcal{I}_m$ | $\mathcal{I}_g^r$ | 0.1 | 22799.0672 | 19.80 | 25.17 | 37.89 | 26.00 | 23.73 | 26.52 |
| 0.1 | $\mathcal{I}_l$ | $\mathcal{I}_g^r$ | 0.1 | 149.9629 | 25.17 | 47.14 | 43.91 | 32.83 | 27.30 | 35.27 |
| 0.1 | $\mathcal{I}_g$ | $\mathcal{I}_l^r$ | 0.1 | 35.4173 | 28.33 | 62.33 | 68.99 | 37.05 | 37.43 | 46.83 |
| 0.1 | $\mathcal{I}_m$ | $\mathcal{I}_l^r$ | 0.1 | 42419.1245 | 21.33 | 26.22 | 37.77 | 25.89 | 24.47 | 27.14 |
| 0.1 | $\mathcal{I}_l$ | $\mathcal{I}_l^r$ | 0.1 | 30996.3493 | 19.97 | 25.21 | 38.17 | 26.05 | 23.10 | 26.50 |
| 0.3 | $\mathcal{I}_g$ | None | 0.0 | 894.2569 | 19.03 | **29.34** | 37.83 | 26.58 | 22.94 | 27.14 |
| 0.3 | $\mathcal{I}_m$ | None | 0.0 | 28379.3026 | 20.48 | 25.97 | 37.95 | 25.97 | 23.91 | 26.86 |
| 0.3 | $\mathcal{I}_l$ | None | 0.0 | 41741.2115 | 20.65 | 24.96 | 37.89 | 25.89 | **25.43** | 26.96 |
| 0.3 | $\mathcal{I}_g$ | $\mathcal{I}_m^r$ | 0.1 | 1031.7398 | 20.05 | 27.53 | 37.83 | 26.62 | 23.18 | 27.04 |
| 0.3 | $\mathcal{I}_l$ | $\mathcal{I}_m^r$ | 0.1 | 30276.3486 | **22.35** | 26.14 | 37.83 | 25.90 | 22.99 | 27.04 |
| 0.3 | $\mathcal{I}_g$ | $\mathcal{I}_g^r$ | 0.1 | **331.4979** | 18.17 | 29.08 | **39.27** | **26.78** | 22.93 | **27.25** |
| 0.3 | $\mathcal{I}_m$ | $\mathcal{I}_g^r$ | 0.1 | 50386.5963 | 19.80 | 24.87 | 37.86 | 25.69 | 22.95 | 26.23 |
| 0.3 | $\mathcal{I}_l$ | $\mathcal{I}_g^r$ | 0.1 | 16962.2387 | 22.18 | 25.17 | 38.01 | 26.09 | 23.10 | 26.91 |
| 0.3 | $\mathcal{I}_g$ | $\mathcal{I}_l^r$ | 0.1 | 618.1265 | 19.20 | 27.78 | 37.83 | 26.71 | 22.95 | 26.89 |
| 0.3 | $\mathcal{I}_m$ | $\mathcal{I}_l^r$ | 0.1 | 32582.2245 | 20.99 | 25.25 | 37.77 | 25.84 | 23.99 | 26.77 |
| 0.3 | $\mathcal{I}_l$ | $\mathcal{I}_l^r$ | 0.1 | 20718.8411 | 18.86 | 26.14 | 37.95 | 26.04 | 23.11 | 26.42 |
| 0.5 | $\mathcal{I}_g$ | None | 0.0 | 8890.5345 | 20.14 | **26.43** | 37.83 | 26.18 | 22.95 | 26.71 |
| 0.5 | $\mathcal{I}_m$ | None | 0.0 | 21843.7764 | 19.37 | 25.84 | **37.92** | 25.89 | **24.95** | 26.79 |
| 0.5 | $\mathcal{I}_l$ | None | 0.0 | 25589.1671 | 19.28 | 25.84 | 37.83 | 26.02 | 23.03 | 26.40 |
| 0.5 | $\mathcal{I}_g$ | $\mathcal{I}_m^r$ | 0.1 | 14315.0669 | 19.54 | 25.93 | 37.83 | 26.15 | 22.94 | 26.48 |
| 0.5 | $\mathcal{I}_l$ | $\mathcal{I}_m^r$ | 0.1 | 19880.3462 | 20.82 | 25.55 | 37.83 | 26.13 | 22.95 | 26.66 |
| 0.5 | $\mathcal{I}_g$ | $\mathcal{I}_g^r$ | 0.1 | 10621.5517 | 18.26 | 26.09 | 37.83 | 26.32 | 22.93 | 26.29 |
| 0.5 | $\mathcal{I}_m$ | $\mathcal{I}_g^r$ | 0.1 | 25005.5409 | **22.10** | 25.97 | 37.83 | 25.85 | 22.95 | 26.94 |
| 0.5 | $\mathcal{I}_l$ | $\mathcal{I}_g^r$ | 0.1 | 17952.1065 | 20.56 | 25.17 | 37.83 | 26.17 | 23.37 | 26.62 |
| 0.5 | $\mathcal{I}_g$ | $\mathcal{I}_l^r$ | 0.1 | **4500.4170** | 18.77 | 25.80 | 37.83 | **26.44** | 22.95 | 26.36 |
| 0.5 | $\mathcal{I}_m$ | $\mathcal{I}_l^r$ | 0.1 | 16028.1470 | 20.56 | 26.05 | 37.83 | 25.82 | 22.95 | 26.64 |
| 0.5 | $\mathcal{I}_l$ | $\mathcal{I}_l^r$ | 0.1 | 17447.9861 | 21.25 | 25.84 | 37.86 | 25.81 | 24.88 | **27.13** |

### A.1.3  2:4

Table 4: Performance for Llama-3.1-8B with 2:4 sparsity.

| Pruning Ratio | Stage I Criterion | Stage II Criterion | Restoring Ratio | WikiText2 | Arc Challenge | Arc Easy | BoolQ | Hellaswag | MMLU | Average |
|---|---|---|---|---|---|---|---|---|---|---|
| 0.5 | $\mathcal{I}_g$ | None | 0.0 | 457.9260 | 19.54 | 31.36 | 38.90 | 26.96 | 22.95 | 27.94 |
| 0.5 | $\mathcal{I}_m$ | None | 0.0 | 151.9501 | 23.72 | 46.84 | 63.21 | 34.81 | 26.36 | 38.99 |
| 0.5 | $\mathcal{I}_l$ | None | 0.0 | **32.9414** | **29.10** | **59.22** | 62.97 | 37.36 | 32.29 | 44.19 |
| 0.5 | $\mathcal{I}_g$ | $\mathcal{I}_m^r$ | 0.2 | 64.5163 | 27.39 | 55.81 | 68.10 | 36.27 | 26.82 | 42.88 |
| 0.5 | $\mathcal{I}_l$ | $\mathcal{I}_m^r$ | 0.2 | 33.8599 | 28.07 | 59.09 | **72.48** | **38.87** | **34.59** | **46.62** |
| 0.5 | $\mathcal{I}_g$ | $\mathcal{I}_g^r$ | 0.2 | 207.7306 | 17.49 | 36.32 | 56.76 | 28.15 | 22.94 | 32.33 |
| 0.5 | $\mathcal{I}_m$ | $\mathcal{I}_g^r$ | 0.2 | 179.3305 | 20.90 | 43.94 | 62.14 | 31.74 | 23.17 | 36.38 |
| 0.5 | $\mathcal{I}_l$ | $\mathcal{I}_g^r$ | 0.2 | 91.5202 | 19.54 | 42.26 | 61.90 | 30.65 | 23.04 | 35.48 |
| 0.5 | $\mathcal{I}_g$ | $\mathcal{I}_l^r$ | 0.2 | 182.0717 | 19.03 | 37.33 | 42.23 | 28.67 | 23.17 | 30.09 |
| 0.5 | $\mathcal{I}_m$ | $\mathcal{I}_l^r$ | 0.2 | 82.5275 | 23.55 | 52.06 | 64.37 | 34.56 | 25.51 | 40.01 |
| 0.5 | $\mathcal{I}_l$ | $\mathcal{I}_l^r$ | 0.2 | 82.3038 | 20.82 | 44.02 | 62.11 | 30.96 | 23.31 | 36.24 |

## A.2 MISTRAL-7B

### A.2.1 UNSTRUCTURED

Table 5: Performance for Mistral-7B with Unstructured sparsity.

| Pruning Ratio | Stage I Criterion | Stage II Criterion | Restoring Ratio | WikiText2 | Arc Challenge | Arc Easy | BoolQ | Hellaswag | MMLU | Average |
|---|---|---|---|---|---|---|---|---|---|---|
| 0.3 | $\mathcal{I}_g$ | None | 0.0 | 27.7559 | 34.56 | 65.82 | 72.84 | 46.61 | 32.79 | 50.52 |
| 0.3 | $\mathcal{I}_m$ | None | 0.0 | 21.5563 | 46.67 | 75.76 | 74.92 | **58.08** | 53.32 | 61.75 |
| 0.3 | $\mathcal{I}_l$ | None | 0.0 | 18.0759 | 46.08 | 75.00 | 80.73 | 56.38 | 54.16 | 62.47 |
| 0.3 | $\mathcal{I}_g$ | $\mathcal{I}_m^r$ | 0.1 | 22.2055 | 44.97 | 72.22 | 79.79 | 54.27 | 54.61 | 61.17 |
| 0.3 | $\mathcal{I}_l$ | $\mathcal{I}_m^r$ | 0.1 | **17.4971** | 45.90 | 75.00 | 80.34 | 56.56 | 55.93 | 62.75 |
| 0.3 | $\mathcal{I}_g$ | $\mathcal{I}_g^r$ | 0.1 | 22.3395 | 39.51 | 70.96 | 63.52 | 50.93 | 43.71 | 53.73 |
| 0.3 | $\mathcal{I}_m$ | $\mathcal{I}_g^r$ | 0.1 | 19.4274 | 46.33 | **76.05** | 74.56 | 56.18 | **56.12** | 61.85 |
| 0.3 | $\mathcal{I}_l$ | $\mathcal{I}_g^r$ | 0.1 | 19.5890 | 44.62 | 75.25 | 80.37 | 55.60 | 55.09 | 62.19 |
| 0.3 | $\mathcal{I}_g$ | $\mathcal{I}_l^r$ | 0.1 | 18.9278 | 44.54 | 75.51 | 77.86 | 55.06 | 53.54 | 61.30 |
| 0.3 | $\mathcal{I}_m$ | $\mathcal{I}_l^r$ | 0.1 | 17.9817 | **47.27** | 75.38 | **80.92** | 56.90 | 55.71 | **63.24** |
| 0.3 | $\mathcal{I}_l$ | $\mathcal{I}_l^r$ | 0.1 | 22.5999 | 44.71 | 73.74 | 80.24 | 55.92 | 55.16 | 61.95 |
| 0.5 | $\mathcal{I}_g$ | None | 0.0 | 232.2574 | 20.65 | 40.40 | 63.33 | 33.34 | 23.64 | 36.27 |
| 0.5 | $\mathcal{I}_m$ | None | 0.0 | 29.3317 | 38.99 | 69.49 | 67.74 | **53.58** | 40.80 | 54.12 |
| 0.5 | $\mathcal{I}_l$ | None | 0.0 | **21.5317** | 41.89 | 71.34 | 77.16 | 51.84 | 39.26 | 56.30 |
| 0.5 | $\mathcal{I}_g$ | $\mathcal{I}_m^r$ | 0.1 | 29.3016 | 36.35 | 67.17 | 74.71 | 48.20 | 37.22 | 52.73 |
| 0.5 | $\mathcal{I}_l$ | $\mathcal{I}_m^r$ | 0.1 | 34.5596 | 37.20 | 62.63 | 75.11 | 44.47 | 47.21 | 53.32 |
| 0.5 | $\mathcal{I}_g$ | $\mathcal{I}_g^r$ | 0.1 | 73.0950 | 22.27 | 45.33 | 62.20 | 33.83 | 25.07 | 37.74 |
| 0.5 | $\mathcal{I}_m$ | $\mathcal{I}_g^r$ | 0.1 | 24.2192 | 39.33 | 69.61 | 59.14 | 50.62 | 44.01 | 52.54 |
| 0.5 | $\mathcal{I}_l$ | $\mathcal{I}_g^r$ | 0.1 | 27.1335 | 36.26 | 67.26 | 74.22 | 47.06 | 39.25 | 52.81 |
| 0.5 | $\mathcal{I}_g$ | $\mathcal{I}_l^r$ | 0.1 | 61.2039 | 25.09 | 51.26 | 60.80 | 35.23 | 25.13 | 39.50 |
| 0.5 | $\mathcal{I}_m$ | $\mathcal{I}_l^r$ | 0.1 | 38.2084 | 40.53 | **71.63** | 76.09 | 51.83 | **47.38** | 57.49 |
| 0.5 | $\mathcal{I}_l$ | $\mathcal{I}_l^r$ | 0.1 | 24.0534 | **42.06** | 71.46 | **78.35** | 50.87 | 45.44 | **57.64** |
| 0.7 | $\mathcal{I}_g$ | None | 0.0 | 673.7895 | 20.22 | 28.87 | 37.86 | 26.35 | 22.94 | 27.25 |
| 0.7 | $\mathcal{I}_m$ | None | 0.0 | 2679.0427 | 21.08 | 37.79 | 38.59 | 31.39 | 23.12 | 30.39 |
| 0.7 | $\mathcal{I}_l$ | None | 0.0 | **109.7167** | 21.84 | **46.46** | 61.56 | 32.19 | **23.50** | 37.11 |
| 0.7 | $\mathcal{I}_g$ | $\mathcal{I}_m^r$ | 0.1 | 364.1544 | 20.48 | 35.44 | 51.47 | 29.63 | 22.95 | 31.99 |
| 0.7 | $\mathcal{I}_l$ | $\mathcal{I}_m^r$ | 0.1 | 527.0593 | **21.93** | 45.45 | **62.48** | **33.06** | 23.12 | **37.21** |
| 0.7 | $\mathcal{I}_g$ | $\mathcal{I}_g^r$ | 0.1 | 612.3245 | 18.34 | 29.55 | 37.86 | 26.70 | 23.12 | 27.11 |
| 0.7 | $\mathcal{I}_m$ | $\mathcal{I}_g^r$ | 0.1 | 2086.6113 | 19.97 | 39.39 | 41.01 | 30.10 | 22.92 | 30.68 |
| 0.7 | $\mathcal{I}_l$ | $\mathcal{I}_g^r$ | 0.1 | 207.3952 | 18.69 | 34.22 | 47.92 | 28.87 | 22.96 | 30.53 |
| 0.7 | $\mathcal{I}_g$ | $\mathcal{I}_l^r$ | 0.1 | 559.7967 | 20.14 | 29.12 | 37.83 | 26.94 | 22.96 | 27.40 |
| 0.7 | $\mathcal{I}_m$ | $\mathcal{I}_l^r$ | 0.1 | 332.1593 | 20.99 | 40.03 | 58.90 | 30.78 | 23.06 | 34.75 |
| 0.7 | $\mathcal{I}_l$ | $\mathcal{I}_l^r$ | 0.1 | 366.4598 | 18.94 | 34.97 | 43.85 | 27.80 | 22.97 | 29.71 |

## A.2.2 CHANNEL-WISE

Table 6: Performance for Mistral-7B with Channel-wise sparsity.

| Pruning Ratio | Stage I Criterion | Stage II Criterion | Restoring Ratio | WikiText2 | Arc Challenge | Arc Easy | BoolQ | Hellaswag | MMLU | Average |
|---|---|---|---|---|---|---|---|---|---|---|
| 0.1 | $\mathcal{I}_g$ | None | 0.0 | 32.7422 | 31.40 | 59.89 | 64.43 | 41.03 | 27.38 | 44.83 |
| 0.1 | $\mathcal{I}_m$ | None | 0.0 | 41827.7809 | 21.08 | 26.52 | 37.83 | 25.89 | 23.78 | 27.02 |
| 0.1 | $\mathcal{I}_l$ | None | 0.0 | 7189.9838 | 21.84 | 28.41 | 37.83 | 26.94 | 24.21 | 27.85 |
| 0.1 | $\mathcal{I}_g$ | $\mathcal{I}_m^r$ | 0.1 | 36.2525 | **34.90** | 63.97 | 66.73 | 43.89 | 31.83 | 48.26 |
| 0.1 | $\mathcal{I}_l$ | $\mathcal{I}_m^r$ | 0.1 | 7645.9814 | 22.18 | 27.10 | 37.83 | 26.76 | 24.36 | 27.65 |
| 0.1 | $\mathcal{I}_g$ | $\mathcal{I}_g^r$ | 0.1 | 38.8391 | 31.91 | **65.15** | 62.69 | 44.17 | **34.55** | 47.69 |
| 0.1 | $\mathcal{I}_m$ | $\mathcal{I}_g^r$ | 0.1 | 2837.7818 | 20.82 | 36.87 | 39.66 | 29.10 | 23.41 | 29.97 |
| 0.1 | $\mathcal{I}_l$ | $\mathcal{I}_g^r$ | 0.1 | 62.7352 | 31.83 | 62.79 | 65.90 | **44.18** | 30.40 | 47.02 |
| 0.1 | $\mathcal{I}_g$ | $\mathcal{I}_l^r$ | 0.1 | **31.9463** | 32.42 | 64.10 | 74.37 | 42.83 | 32.89 | **49.32** |
| 0.1 | $\mathcal{I}_m$ | $\mathcal{I}_l^r$ | 0.1 | 40875.1467 | 22.10 | 26.81 | 37.83 | 25.93 | 23.79 | 27.29 |
| 0.1 | $\mathcal{I}_l$ | $\mathcal{I}_l^r$ | 0.1 | 1749.7535 | 22.87 | 32.37 | 38.13 | 28.72 | 23.75 | 29.17 |
| 0.3 | $\mathcal{I}_g$ | None | 0.0 | 301.0231 | 19.62 | 29.88 | 37.89 | 26.62 | 22.95 | 27.39 |
| 0.3 | $\mathcal{I}_m$ | None | 0.0 | 32444.2689 | 20.90 | 27.10 | 37.83 | 25.97 | 23.02 | 26.96 |
| 0.3 | $\mathcal{I}_l$ | None | 0.0 | 43157.3757 | **21.50** | 24.92 | 37.83 | 25.84 | 23.54 | 26.73 |
| 0.3 | $\mathcal{I}_g$ | $\mathcal{I}_m^r$ | 0.1 | 581.7826 | 19.97 | 30.22 | 40.24 | 26.93 | 22.95 | 28.06 |
| 0.3 | $\mathcal{I}_l$ | $\mathcal{I}_m^r$ | 0.1 | 57163.4627 | **21.50** | 26.52 | 37.83 | 25.97 | 23.44 | 27.05 |
| 0.3 | $\mathcal{I}_g$ | $\mathcal{I}_g^r$ | 0.1 | **202.4535** | 18.94 | **30.43** | 58.32 | **27.53** | 22.96 | **31.64** |
| 0.3 | $\mathcal{I}_m$ | $\mathcal{I}_g^r$ | 0.1 | 67489.6705 | 20.82 | 25.76 | 37.83 | 26.19 | 23.00 | 26.72 |
| 0.3 | $\mathcal{I}_l$ | $\mathcal{I}_g^r$ | 0.1 | 5705.7535 | 19.37 | 29.25 | 45.41 | 26.39 | 22.97 | 28.68 |
| 0.3 | $\mathcal{I}_g$ | $\mathcal{I}_l^r$ | 0.1 | 317.6567 | 18.94 | 29.67 | 40.49 | 27.01 | 22.93 | 27.81 |
| 0.3 | $\mathcal{I}_m$ | $\mathcal{I}_l^r$ | 0.1 | 51640.4425 | 21.42 | 26.14 | 37.83 | 25.80 | 22.82 | 26.80 |
| 0.3 | $\mathcal{I}_l$ | $\mathcal{I}_l^r$ | 0.1 | 37984.1039 | 20.99 | 26.05 | 37.83 | 25.65 | **23.75** | 26.85 |
| 0.5 | $\mathcal{I}_g$ | None | 0.0 | 1354.7247 | 20.56 | **27.86** | 37.83 | 26.07 | 22.94 | 27.05 |
| 0.5 | $\mathcal{I}_m$ | None | 0.0 | 29759.1804 | 21.76 | 26.89 | 37.83 | 25.89 | 23.03 | 27.08 |
| 0.5 | $\mathcal{I}_l$ | None | 0.0 | 38570.4604 | 20.82 | 27.19 | 37.83 | 26.05 | 22.95 | 26.97 |
| 0.5 | $\mathcal{I}_g$ | $\mathcal{I}_m^r$ | 0.1 | 2426.7262 | 20.05 | 27.19 | 37.83 | 26.00 | 22.94 | 26.80 |
| 0.5 | $\mathcal{I}_l$ | $\mathcal{I}_m^r$ | 0.1 | 30946.2993 | 21.67 | 26.73 | 37.83 | 26.06 | 25.17 | **27.49** |
| 0.5 | $\mathcal{I}_g$ | $\mathcal{I}_g^r$ | 0.1 | **1089.5944** | 19.97 | 27.44 | 37.83 | **26.42** | 22.95 | 26.92 |
| 0.5 | $\mathcal{I}_m$ | $\mathcal{I}_g^r$ | 0.1 | 38273.0667 | 21.16 | 24.83 | 37.83 | 26.14 | 24.28 | 26.85 |
| 0.5 | $\mathcal{I}_l$ | $\mathcal{I}_g^r$ | 0.1 | 15305.4860 | 20.05 | 26.01 | 37.77 | 25.89 | 22.87 | 26.52 |
| 0.5 | $\mathcal{I}_g$ | $\mathcal{I}_l^r$ | 0.1 | 1803.6687 | 19.71 | 26.64 | 37.83 | 26.38 | 23.00 | 26.71 |
| 0.5 | $\mathcal{I}_m$ | $\mathcal{I}_l^r$ | 0.1 | 29890.9204 | 19.97 | 26.68 | 37.83 | 25.92 | **25.41** | 27.16 |
| 0.5 | $\mathcal{I}_l$ | $\mathcal{I}_l^r$ | 0.1 | 38724.1943 | **22.44** | 26.64 | 37.71 | 26.03 | 22.93 | 27.15 |

