# OpenReview forum: "Posterior Restoration for Enhanced LLM Pruning"
_ICLR.cc/2026/Conference — Submitted to ICLR 2026_

### Official Review · Reviewer_qizh · 2025-10-20

**Soundness:** 1
**Presentation:** 2
**Contribution:** 2
**Rating:** 2
**Confidence:** 4

**Summary:**

This paper addresses two limitations in existing post-training pruning algorithms for large language models (LLMs): (1) overfitting on calibration data and (2) the neglect of weight interactions during pruning. To tackle these issues, the authors propose a two-stage pruning framework. In the first stage, a prior criterion is used to generate a coarse pruning mask. In the second stage, posterior criteria are introduced to restore important weights that were mistakenly pruned. Three types of posterior criteria are designed—magnitude-based, local, and global—which aim to capture different levels of weight importance and interactions. Experiments on LLaMA-3.1-8B and Mistral-7B models demonstrate modest perplexity improvements compared to standard post-training pruning baselines, particularly at moderate sparsity levels.

However, I find that neither the theoretical analysis nor the experimental results provide strong support for the authors’ claims, and therefore I lean to reject this paper.

**Strengths:**

Originality:
The paper explores a two-stage framework combining prior and posterior criteria for LLM pruning, which is a novel conceptual attempt to improve post-training pruning without full retraining. The idea of “posterior restoration” provides an interesting perspective on mitigating pruning errors.

Quality:
The implementation covers multiple pruning granularities (structured, semi-structured, and unstructured) and evaluates across two modern LLMs (LLaMA-3.1-8B and Mistral-7B). The framework is clearly described with explicit definitions of the three posterior criteria.

Clarity:
The paper is overall well-structured, clearly separating theoretical motivation, method design, and experimental validation. The figures and tables are easy to follow.

Significance:
The work tackles an important topic in LLMs pruning refinement. Even though the improvements are small, the exploration of posterior-based recovery may offer an alternative approach to performance recovery beyond iterative and trainable pruning methods.

**Weaknesses:**

1. Theoretical inconsistency:
The authors argue that current post-training pruning methods risk overfitting to calibration data and perform poorly when input variance is large, as well as ignore weight interactions. Yet, their proposed approach still relies on these same methods to generate the coarse mask in the first stage, inheriting the same weaknesses. If the posterior criterion truly mitigates these issues, it raises the question of why it is not applied directly from the beginning. This makes the proposed two-stage design appear self-contradictory and unnecessary.

2. Unconvincing experimental evidence:
(1) The posterior global criterion only works on structured pruning. Although the authors claim that it captures interactions among large blocks of weights, the experimental results do not support this claim. In unstructured and semi-structured pruning, the same issue of neglecting weight interactions exists. Since the performance becomes worse when applying the posterior global criterion to these two settings, it indicates that the proposed method is ineffective.
(2) The posterior local criterion combined with the magnitude prior is claimed to be most effective at 30% sparsity, however the perplexity only improves marginally (from 11.8 to 11.6), which seems not to be statistically significant.
(3) The paper targets post-training pruning, but the authors employ LoRA fine-tuning. Pruning can be evaluated directly without additional training, so this design choice is unclear.

3. Limited scale and generality:
Experiments are only conducted on 7B-parameter models, lacking evidence on larger models (e.g., 13B–70B), which limits the generalizability of conclusions.

**Questions:**

1. Conceptual clarification:
If the posterior criteria are designed to mitigate overfitting and capture weight interactions, why are they not used directly to generate the pruning mask? What is the fundamental necessity of the first (prior) stage?

2. Experimental design:
(1) Why introduce LoRA fine-tuning in what is described as a “post-training pruning” setting? Could the authors report results without fine-tuning to isolate pruning effectiveness?
(2) Can the authors include larger models (e.g., LLaMA-13B or 70B) to demonstrate scalability?

3. Evaluation and significance:
(1) Are the perplexity differences statistically significant? Confidence intervals or multiple runs could clarify the robustness of improvements.
(2) Could the authors provide qualitative or sensitivity analyses showing how posterior restoration changes the sparsity pattern or weight distributions?

---

### Official Review · Reviewer_c9LL · 2025-10-23

**Soundness:** 3
**Presentation:** 3
**Contribution:** 2
**Rating:** 2
**Confidence:** 4

**Summary:**

This paper introduces Posterior Restoration, a two-step framework designed to enhance LLM pruning by reducing overfitting and capturing weight interactions ignored by traditional prior-based pruning methods.

The research question is: _How can the importance of weights be better estimated by considering post-pruning information instead of relying only on dense-model priors?_

**Strengths:**

- The paper is well written, with smooth flow and clear sectioning.

- The idea is somehow novelty and can contribute to the community.

- Figures 2 and 3 clearly illustrate the effect of the restoring ratio $\Delta p$ and visualize the refinement process, helping readers grasp the intuition behind posterior restoration.

**Weaknesses:**

- No baseline
- Lack of experiments showing the cost and pruning time
- Lack of experiments showing the efficiency acceleration
- Two models in comparison is insufficiency. Need scalability test.
- It is a unfinished work.

**Questions:**

- How sensitive is posterior restoration to the choice of $\Delta p$ across different sparsity regimes?
- Can posterior global and local criteria be jointly optimized within a single stage?
- What is the computational cost (in GPU-hours or FLOPs) of computing posterior gradients on pruned models?

---

### Official Review · Reviewer_JhPA · 2025-10-27

**Soundness:** 2
**Presentation:** 2
**Contribution:** 1
**Rating:** 2
**Confidence:** 4

**Summary:**

This paper proposes posterior restoration, a simple two-stage approach for LLM pruning. First, it applies a conventional prior criterion to generate an initial coarse pruning mask. Second, it restores the most important weights, guided by our novel posterior criteria (magnitude, global, and local), which re-evaluate the removed weights from the perspective of the already-pruned model. It presents three types of posterior criteria: magnitude, global, and local, to address the limitations of prior criteria.

**Strengths:**

It can seamlessly integrate and enhance most existing pruning methods.

**Weaknesses:**

The novelty of the proposed method may be limited. The two stage method mainly prune the mask first and then update the mask. This is very common in pruning. Some method can update the mask multiple times such as sparsegpt. The three posterior criterias are simple and mainly follows previous works, such as magnitude and Wanda. The technical contribution may be limited.

The experiments can be enhanced. It only experiments with 7B size models. It is better to experiment with other model sizes to demonstrate the general performance.

It mentions to use global criterion to capture the weight interactions that prior methods ignore by reversing the perspective of the Taylor expansion. But the weight interactions are investigated with the loss from previous work  Molchanov et al. (2019). This paper does not make specific contributions to further investigate weight interactions. similarly,  the layer’s output error the preceding pruned layer is investigated with posterior local criterion following previous work Wanda. This paper does not make specific contributions to further investigate  error accumulation. The contributions may be limited.

It claims that the posterior magnitude criterion mitigates the overfitting introduced by data-dependent prior methods. But this paper does not provide solid discussions or experiments to demonstrate why  posterior magnitude criterion can mitigate the overfitting. It is hard to see the connection. It is better to provide more  discussions and experiments.

It is better to discuss the complexity of the proposed method.

The baselines are weak. There are almost no baselines in this paper. This paper is a general pruning work. It can compare with other pruning methods. It is better to compare with more baselines. For example, the ppl under 50% unstructured sparsity of SparseGPT degrades very slightly. But in this paper, the PPL under 50% unstructured sparsity drops from 8.64 to 16.7. It seems that the performance is not competitive.

It only finds a pruning mask. But in practice, models are typically finetuned to achieve better performance. It is hard to say whether the mask is good enough before actual finetuning. It does not seem to make sense to just compare the ppl or accuracy under un-finetuned mask. People probably do not use this model since they are not finetuned with worse performance. It is better to find other methods to compare the performance of different masks.

**Questions:**

See the weakness.

---

### Official Review · Reviewer_HtKa · 2025-10-29

**Soundness:** 3
**Presentation:** 3
**Contribution:** 2
**Rating:** 2
**Confidence:** 4

**Summary:**

The paper introduces "posterior restoration," a two-stage pruning method for large language models (LLMs). In the first stage, a conventional prior criterion (magnitude, global, or local) is used to generate a coarse pruning mask at a higher sparsity level than desired. In the second stage, pruned weights are re-evaluated and partially restored using novel "posterior" criteria, which aim to mitigate overfitting to calibration data and capture weight interactions ignored by prior methods.

**Strengths:**

1. The paper provides a clear algorithmic outline and visual explanations, which help illustrate the method.

**Weaknesses:**

1. The scope of experiments is limited. The author should compare the proposed method with different pruning methods in the first stage, such as Wanda and SparseGPT.
2. The method relies on a stack of assumptions, many of which are questionable. For instance, Assumption 1 (independence of pruning errors) is not supported by any experimental evidence.
3. The posterior restoration phase disrupts the predefined sparsity structure in semi-structured pruning methods, such as 2:4 sparsity, effectively transforming it into an unstructured sparsity pattern that lacks hardware optimization compatibility.
4. The related work is sufficient. There are other refinement methods that also mitigate the gap between a dense model and a pruned model. For example, low-rank refinement [1, 2] and adaptive layer-wise sparsity control [3].

[1] Y. Li et al., “LoSparse: Structured Compression of Large Language Models based on Low-Rank and Sparse Approximation,” 2023.
[2] L. Shen, A. Tang, Y. Luo, T. Sun, H. Hu, and X. Cao, “Targeted Low-rank Refinement: Enhancing Sparse Language Model with Precision,” 2025.
[3] L. Yin et al., “Outlier Weighed Layerwise Sparsity (OWL): A Missing Secret Sauce for Pruning LLMs to High Sparsity,” 2024.

**Questions:**

1. How sensitive is performance to $\Delta p$? Provide an ablation.

---

### Meta-Review · Area_Chair_Vz3s · 2026-01-01

**Summary:**

This paper presents a two-stage pruning method for large language models, addressing the limitations of existing approaches, such as overfitting to calibration data and disregarding weight interactions. The primary concept involves using a conventional prior criterion to generate an initial coarse pruning mask, followed by restoring the most important weights using a posterior criterion extended from existing methods. However, significant concerns remain regarding the novelty of the proposed approach and the adequacy of the experimental evaluations. Additionally, the authors did not submit any rebuttal materials. As a result, the paper is not recommended for acceptance.

**Reviewer Concerns:**

The authors did not submit any rebuttal materials, therefore, none concerns have been addredded by the authors.

Outstanding Concerns:

**1. The experimental evaluation is insufficient. (Reviewer HtKa, JhPA,c9LL, qizh).** For example, the author should compare the proposed method with different pruning methods, conduct experimets with more LLMs.

**2. The novelty of the proposed method is weak. (Reviewer JhPA).** . The two stage pruning schedule, which  prunes the mask first and then update the mask, is very common. The key techniques, i.e., the three posterior criterias, are simple and mainly follows previous works, such as magnitude and Wanda.

**Reviewer Scores:**

All reviewers gave this paper a rating of 2 and did not revise their scores, as the aurhors did not submit the rebuttal.

---

### Decision · Program_Chairs · 2026-01-26

Reject